# Primate phylogenomics uncovers multiple rapid radiations and ancient interspecific introgression

Dan Vanderpool[1]*, Bui Quang Minh[2,3], Robert Lanfear[3], Daniel Hughes[4],
Shwetha Murali[4], R. Alan Harris[4,5], Muthuswamy Raveendran[4], Donna M. Muzny[4,5],
Mark S. Hibbins[1], Robert J. Williamson[6], Richard A. Gibbs[4,5], Kim C. Worley[4,5],
Jeffrey Rogers[4,5], Matthew W. Hahn[1]

1 Department of Biology and Department of Computer Science, Indiana University, Bloomington, Indiana,
United States of America, 2 Research School of Computer Science, Australian National University,
Canberra, Australian Capital Territory, Australia, 3 Research School of Biology, Australian National
University, Canberra, Australian Capital Territory, Australia, 4 Human Genome Sequencing Center, Baylor
College of Medicine, Houston, Texas, United States of America, 5 Department of Molecular and Human
Genetics, Baylor College of Medicine, Houston, Texas, United States of America, 6 Department of Computer
Science and Software Engineering and Department of Biology and Biomedical Engineering, Rose-Hulman
Institute of Technology, Terre Haute, Indiana, United States of America

* danvand@indiana.edu

Cambridge, UNITED KINGDOM

**Data Availability Statement:** The relevant
assembly accessions and associated references
used in this study are provided in S1 Table. All raw
data, assemblies, and annotation information used

## Abstract

Our understanding of the evolutionary history of primates is undergoing continual revision
due to ongoing genome sequencing efforts. Bolstered by growing fossil evidence, these
data have led to increased acceptance of once controversial hypotheses regarding phyloge-
netic relationships, hybridization and introgression, and the biogeographical history of pri-
mate groups. Among these findings is a pattern of recent introgression between species
within all major primate groups examined to date, though little is known about introgression
deeper in time. To address this and other phylogenetic questions, here, we present new ref-
erence genome assemblies for 3 Old World monkey (OWM) species: *Colobus angolensis*
ssp. *palliatus* (the black and white colobus), *Macaca nemestrina* (southern pig-tailed
macaque), and *Mandrillus leucophaeus* (the drill). We combine these data with 23 additional
primate genomes to estimate both the species tree and individual gene trees using thou-
sands of loci. While our species tree is largely consistent with previous phylogenetic hypoth-
eses, the gene trees reveal high levels of genealogical discordance associated with multiple
primate radiations. We use strongly asymmetric patterns of gene tree discordance around
specific branches to identify multiple instances of introgression between ancestral primate
lineages. In addition, we exploit recent fossil evidence to perform fossil-calibrated molecular
dating analyses across the tree. Taken together, our genome-wide data help to resolve mul-
tiple contentious sets of relationships among primates, while also providing insight into the
biological processes and technical artifacts that led to the disagreements in the first place.

in theses analyses are available through each species' NCBI BioProject link available in the relevant assembly accessions and associated references used in this study are provided in S1 Table. The Data Dryad repository associated with this study can be accessed through the following link: https://doi.org/10.5061/dryad.rfj6q577d [22]. The repository contains the following files and archives: • 1730_ALIGNMENT_CONCAT.paup.nex – Concatenated alignment with PAUP block of commands used to generate the parsimony concatenated tree. • 1730_Alignments_FINAL.tar. gz: 1,730 single-copy ortholog alignments. • 1730_ML_GENETREEs.treefile: Maximum likelihood gene trees estimated from from the 1,730 ortholog alignments. • ASTRAL_Tree_AVGdates.tre: The ASTRAL topology (Fig 1) with average dates from 10 independent datasets. • All_Dating_Datasets_DRYAD.tar.gz: The concatenated alignments used for dating analyses. • PARSIMONY_1730_Gene_Trees.tre: All 1,730 parsimony gene trees fro MPBoot. • Supp_fig4A_F_s6_b1_v2_1.slim-SLiM3 recipe for S4A Fig simulation • Supp_fig4B_F_s6_b1_v2_highmut_1.slim- SLiM3 recipe for S4B Fig simulation • All_1735_UNALIGNED_Seqs.tar.gz: All unaligned single-copy gene sequences. • plot_DATING.R: R script used for plotting Fig 3.

**Funding:** Funding for this study was provided by grants from the National Science Foundation, grant numbers: DBI-1564611 and DEB-1936187 awarded to M.W.H. Salary was provided to D.V. and M.W.H by grant number: DBI-1564611. Additional salary was provided to M.W.H by grant number: DEB-1936187. The authors received no specific funding for this work. Additional funding was provided by the Chan-Zuckerberg Initiative grant for Essential Open Source Software for Science (https://chanzuckerberg.com/eoss/) awarded to B.Q.M. and R.L. The authors received no specific funding for this work. Additional funding was provided by the Australian Research Council under grant number: DP-200103151 awarded to R.L., B.Q.M., and M.W.H. The authors received no specific funding for this work. Additional funding was provided by a Australian National University (https://www.anu.edu.au/) Futures grant awarded to R.L, which paid salary for B.Q.M. The sequencing and assembly of the colobus, pig-tailed macaque, and drill genomes was funded by National Institutes of Health grant number: U54-HG006484 awarded to R.G. Salaries from this award were received by R.G., J.R., K.W., S.M., D.H., and D.M. The funders had no role in

## Introduction

Understanding the history of individual genes and whole genomes is an important goal for evolutionary biology. It is only by understanding these histories that we can understand the origin and evolution of traits—whether morphological, behavioral, or biochemical. Until recently, our ability to address the history of genes and genomes was limited by the availability of comparative genomic data. However, genome sequences are now being generated extremely rapidly. In primates alone, there are already 23 species with published reference genome sequences and associated annotations (S1 Table), as well as multiple species with population samples of whole genomes [1–11]. These data can now be used to address important evolutionary questions.

Several studies employing dozens of loci sampled across broad taxonomic groups have provided rough outlines of the evolutionary relationships and divergence times among primates [12,13]. Due to the rapid nature of several independent radiations within primates, these limited data cannot resolve species relationships within some clades [12–14]. For instance, the New World monkeys (NWM) experienced a rapid period of diversification approximately 15 to 18 million years ago (mya) [15] (Fig 1), resulting in ambiguous relationships among the 3 Cebidae subfamilies (Cebinae = squirrel monkeys and capuchins, Aotinae = owl monkeys, and Callitrichinae = marmosets and tamarins) [12–14,16–18]. High levels of incomplete lineage sorting (ILS) driven by short times between the divergence of distinct lineages have led to a large amount of gene tree discordance in the NWM, with different loci favoring differing relationships among taxa. Given the known difficulties associated with resolving short internodes [19–21], as well as the multiple different approaches and datasets used in these analyses, the relationships among cebid subfamilies remain uncertain.

In addition to issues of limited data and rapid radiations, a history of hybridization and subsequent gene flow between taxa means that there is no single dichotomously branching tree that all genes follow. Although introgression once was thought to be relatively rare (especially among animals [23]), genomic studies have uncovered widespread patterns of recent introgression across the tree of life [24]. Evidence for recent or ongoing gene flow is especially common among the primates (e.g., [9,25–27]), sometimes with clear evidence for adaptive introgression (e.g., [28–30]). Whether widespread gene flow among primates is emblematic of their initial radiation (which began 60 to 75 mya [13,31–33]) or is a consequence of current conditions—which include higher environmental occupancy and more secondary contact—remains an open question [34].

Here, we report the sequencing and annotation of 3 new primate genomes, all Old World monkey (OWM) species: *Colobus angolensis* ssp. *palliatus* (the black and white colobus), *Macaca nemestrina* (southern pig-tailed macaque), and *Mandrillus leucophaeus* (the drill). Together with the published whole genomes of extant primates, we present a phylogenomic analysis including 26 primate species and several closely related non-primates. Incorporating recently discovered fossil evidence [35], we perform fossil-calibrated molecular dating analyses to estimate divergence times, including dates for the crown primates as well as the timing of more recent splits. Compared to recent hybridization, introgression that occurred between 2 or more ancestral lineages (represented by internal branches on a phylogeny) is difficult to detect. To get around this limitation, we modify a previously proposed method for detecting introgression [36] and apply it to our whole-genome datasets, finding additional evidence for gene flow among ancestral primates. Finally, we closely examine the genealogical patterns left behind by the NWM radiation, as well as the biases of several methods that have been used to resolve this topology. We use multiple approaches to provide a strongly supported history of

study design, data collection and analysis, decision to publish, or preparation of the manuscript.

**Competing interests:** The authors have declared that no competing interests exist.

**Abbreviations:** BUSCO, Benchmarking Universal Single-Copy Orthologs; CDS, coding sequences; gCF, gene concordance factor; HGCS, Human Genome Sequencing Center; ILS, incomplete lineage sorting; ML-CONCAT, maximum likelihood concatenated; my, million years; mya, million years ago; NWM, New World monkey; OWM, Old World monkey; PAUP*, Phylogenetic Analysis Using Parsimony*; sCF, site concordance factor; SRA, Short Read Archive.

the NWM and primates in general, while also highlighting the large amounts of gene tree discordance across the tree caused by ILS and introgression.

## Results and discussion

### Primate genome sequencing

The 3 species sequenced here are all OWMs, and each is closely related to an already-sequenced species. This sampling scheme provides us increased power to detect introgression among each of the sub-clades containing these species. The assembly and annotation of each of the 3 species sequenced for this project are summarized here, with further details listed in Table 1. A summary of all published genomes used in this study, including links to the assemblies and NCBI BioProjects, is available in S2 Table. All species were sequenced using standard methods according to Illumina (San Diego, California, United States of America) Hi-seq protocols. Additional long-read sequencing was performed using Pacific Biosciences (Menlo Park, California, USA) technology for *M. nemestrina*.

The sequencing effort for *C. angolensis* ssp. *palliatus* produced 514 Gb of data, which are available in the NCBI Short Read Archive (SRA) under the accession SRP050426 (BioProject PRJNA251421). The biological sample used for sequencing was kindly provided by Dr. Oliver Ryder (San Diego Zoo). Assembly of these data resulted in a total assembly length of 2.97 Gb in 13,124 scaffolds (NCBI assembly Cang.pa_1.0; GenBank accession GCF_000951035.1) with an average per base coverage of 86.8X. Subsequent annotation via the NCBI Eukaryotic Genome Annotation Pipeline (annotation release ID: 100) resulted in the identification of 20,222 protein-coding genes and 2,244 noncoding genes. An assessment of the annotation performed using Benchmarking Universal Single-Copy Orthologs (BUSCO) 3.0.2 [37] in conjunction with the Euarchontoglires ortholog database 9 (https://busco-archive.ezlab.org/v3/datasets/euarchontoglires_odb9.tar.gz) indicated that 95.82% single-copy orthologs (91.68% complete and 4.13% fragmented) were present among the annotated protein-coding genes. Comprehensive annotation statistics for *C. angolensis* ssp. *palliatus* with links to the relevant annotation products available for download can be viewed at https://www.ncbi.nlm.nih.gov/genome/annotation_euk/Colobus_angolensis_palliatus/100/.

For *M. nemestrina*, 1,271 Gb of data were produced (SRA accession SRP045960; BioProject PRJNA251427), resulting in an assembled genome length of 2.95 Gb in 9,733 scaffolds (Mnem_1.0; GenBank accession GCF_000956065.1). This corresponds to an average per base coverage of 113.1X when both short- and long-read data are combined (Materials and methods). The biological sample used for sequencing was kindly provided by Drs. Betsy Ferguson and James Ha (Washington National Primate Research Center). The NCBI annotation resulted in 21,017 protein-coding genes and 13,163 noncoding genes (annotation release ID: 101). A BUSCO run to assess the completeness of the annotation (as above) indicated that 95.98% single-copy orthologs (92.23% complete and 3.75% fragmented) were present among the annotated protein-coding genes. Comprehensive annotation statistics for *M. nemestrina* with links to the relevant annotation products available for download can be viewed at https://www.ncbi.nlm.nih.gov/genome/annotation_euk/Macaca_nemestrina/101/.

Sequencing of *M. leucophaeus* libraries resulted in 334.1 Gb of data (SRA accession SRP050495; BioProject PRJNA251423) that once assembled resulted in a total assembly length of 3.06 Gb in 12,821 scaffolds (Mleu.le_1.0; GenBank accession GCF_000951045.1) with an average coverage of 117.2X per base. The biological sample used for sequencing was kindly provided by Dr. Oliver Ryder (San Diego Zoo). The NCBI annotation produced 20,465 protein-coding genes and 2,300 noncoding genes (annotation release ID: 100). A BUSCO run to assess the completeness of the annotation (as above) indicated that 95.45% single-copy

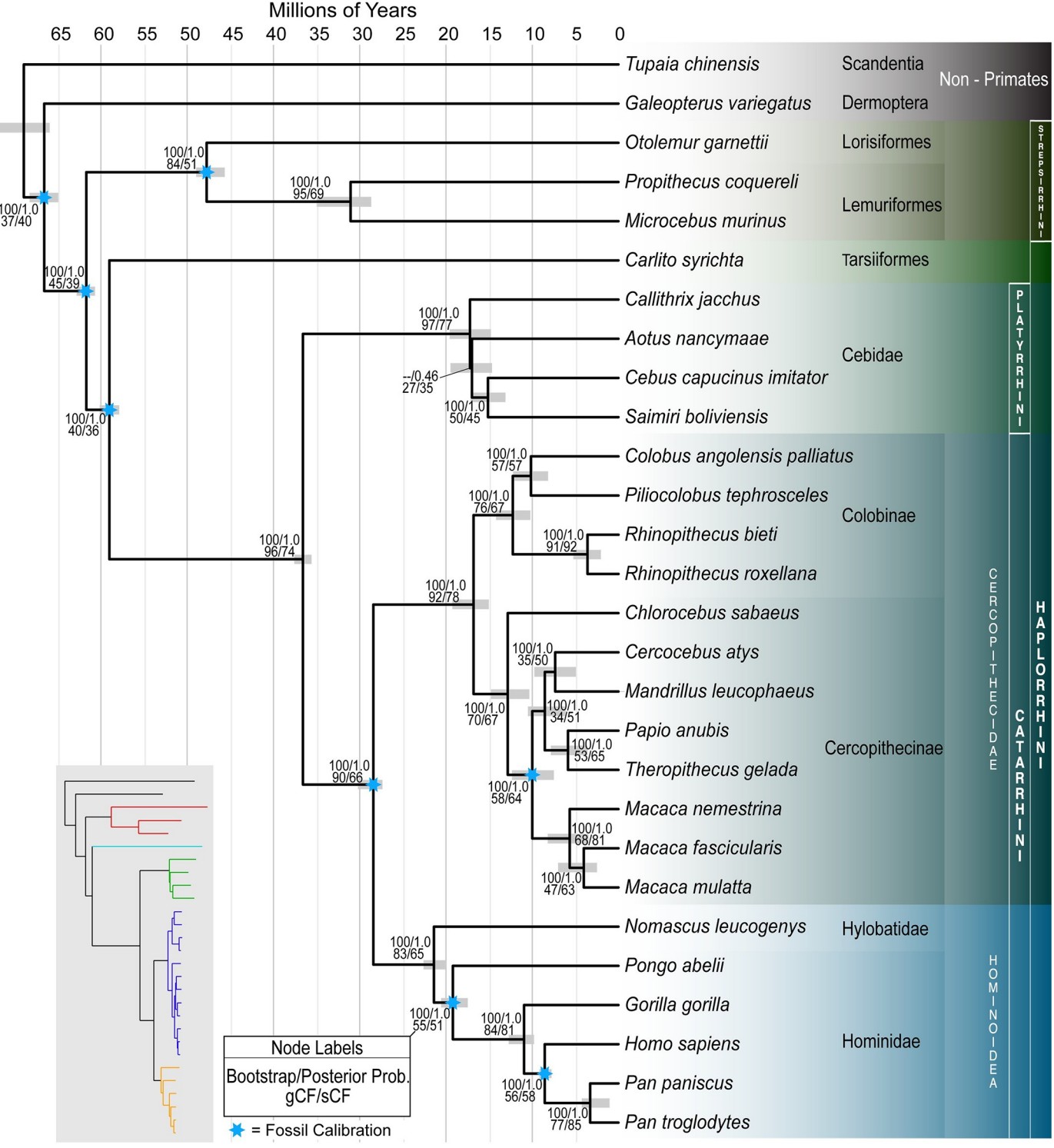

**Fig 1. Species tree estimated using ASTRAL III with 1,730 gene trees (the *Mus musculus* outgroup was removed to allow for a visually finer scale).** Common names for each species can be found in S1 Table. Node labels indicate the bootstrap value from a maximum likelihood analysis of the concatenated dataset as well as the local posterior probability from the ASTRAL analysis. gCFs and sCFs are also reported. Eight fossil calibrations (blue stars; S6 Table) were used to calibrate node ages. Gray bars indicate the minimum and maximum mean age from independent dating estimates. The inset tree with colored branches shows the maximum likelihood branch lengths estimated using a partitioned analysis of the concatenated alignment. Colors correspond to red = Strepsirrhini, cyan = Tarsiiformes, green = Platyrrhini (NWMs), blue = Cercopithecoidea (OWMs), and orange = Hominoidea (Apes). All alignments used for phylogenomic analyses (1730_Alignments_FINAL.tar.gz) and dating analyses (All_Dating_Datasets_DRYAD.tar.gz) are available via Data Dryad: https://doi.org/10.5061/dryad.rfj6q577d [22]. gCF, gene concordance factor; NWM, New World monkey; OWM, Old World monkey; sCF, site concordance factor.

**Table 1. Genomes sequenced in this study and associated assembly and annotation metrics.**

| Species name | Assembly accession | Assembly total length | No. of scaffolds | Scaffold N50 (mb) | Contig N50 (kb) | Protein-coding genes | BUSCO |
|---|---|---|---|---|---|---|---|
| *Colobus angolensis* ssp. *palliatus* (the black and white colobus) | GCF_000951035.1 | 2,970,124,662 | 13,124 | 7.84 | 38.36 | 20,222 | 95.82% |
| *Macaca nemestrina* (pig-tailed macaque) | GCF_000956065.1 | 2,948,703,511 | 9,733 | 15.22 | 106.89 | 21,017 | 95.98% |
| *Mandrillus leucophaeus* (drill) | GCF_000951045.1 | 3,061,992,840 | 12,821 | 3.19 | 31.35 | 20,465 | 95.45% |

BUSCO percentages reflect the complete and fragmented genes relative to the Euarchontoglires ortholog database v9.

BUSCO, Benchmarking Universal Single-Copy Orthologs.

orthologs (91.38% complete, 4.07% fragmented) were present among the annotated protein-coding genes. The full annotation statistics with links to the associated data can be viewed at https://www.ncbi.nlm.nih.gov/genome/annotation_euk/Mandrillus_leucophaeus/100/.

## Phylogenetic relationships among primates

To investigate phylogenetic relationships among primates, we selected the longest isoform for each protein-coding gene from 26 primate species and 3 non-primate species (S1 Table). After clustering, aligning, trimming, and filtering (Materials and methods), there were 1,730 single-copy orthologs present in at least 27 of the 29 species (see S3 Table for the orthogroup, protein name, chromosome, and location of each single-copy ortholog in the human genome). The cutoffs used to filter the dataset ensure high species coverage while still retaining a large number of orthologs. The coding sequences (CDS) of these orthologs have an average length of 1,018 bp and 178 parsimony informative characters per gene. Concatenation of these loci resulted in an alignment of 1,761,114 bp, with the fraction of gaps/ambiguities varying from 4.04% (*Macaca mulatta*) to 18.37% (*Carlito syrichta*) (S4 Table).

We inferred 1,730 individual gene trees from nucleotide alignments using maximum likelihood in IQ-TREE 2 [38] and then inferred a species tree using these gene tree topologies as input to ASTRAL III ([39]; Materials and methods). We used the mouse, *Mus musculus*, as an outgroup to root the species tree. This approach resulted in a topology (which we refer to as "ML-ASTRAL"; Fig 1) that largely agrees with previously published phylogenies [12,13]. We also used IQ-TREE to carry out a maximum likelihood analysis of the concatenated nucleotide alignment (a topology we refer to as ML-CONCAT). This analysis resulted in a topology that differed from the ML-ASTRAL tree only with respect to the placement of *Aotus nancymaae* (owl monkey), rather than sister to the *Saimiri*+*Cebus* clade (as in Fig 1), the ML-CONCAT tree places *Aotus* sister to *Callithrix jacchus*, a minor rearrangement around a very short internal branch (Fig 1). All branches of the ML-ASTRAL species tree are supported by maximum local posteriors, the default support values provided by ASTRAL III [40], except for the branch that defines *Aotus* as sister to the *Saimiri*+*Cebus* clade (0.46 local posterior probability). Likewise, each branch in the ML-CONCAT tree is supported by 100% bootstrap values, including the branch uniting *Aotus* and Callithrix. We return to this conflict in the next section.

There has been some contention as to the placement of the mammalian orders Scandentia (treeshrews) and Dermoptera (colugos) [41–50]. The controversy concerns whether Dermoptera is sister to Primates, Scandentia is sister to Primates, or Dermoptera and Scandentia are sister groups. As expected, both the ML-ASTRAL and ML-CONCAT trees place these 2 groups outside the Primates with maximal statistical support (i.e., local posterior probabilities of 1.0 and bootstrap values of 100%; Fig 1); they also both point to Dermoptera as the closest sister lineage to the Primates [12,51–53]. However, while support values such as the bootstrap

or posterior probability provide statistical confidence in the species tree topology, there can be large amounts of underlying gene tree discordance even for branches with 100% support (e.g., [54–56]). To assess discordance generally, and the relationships among the Primates, Scandentia, and Dermoptera in particular, we used IQ-TREE to calculate both the gene concordance factor (gCF) and site concordance factor (sCF) [57] for each internal branch of the topology in Fig 1. These 2 measures represent the fraction of genes and sites, respectively, which are in agreement with the species tree for any particular branch.

Examining concordance factors helps to explain previous uncertainty in the relationships among Primates, Scandentia, and Dermoptera (Fig 1). Although the bootstrap support is 100% and the posterior probability is 1.0 on the branch leading to the Primate common ancestor, the gCF is 45%, and the sCF is 39%. These values indicate that, of decisive gene trees ($n = 1,663$), only 45% of them contain the branch that is in the species tree; this branch reflects the Primates as a single clade that excludes Scandentia and Dermoptera. While the species tree represents the single topology supported by the most gene trees (hence the strong statistical support for this branch), the concordance factors also indicate that a majority of gene tree topologies differ from the estimated species tree. In fact, the gCF value indicates that 55% of trees do not support a monophyletic Primate order, with either Dermoptera, Scandentia, or both lineages placed within Primates. Likewise, the sCF value indicates that only 39% of decisive sites in the total alignment support the branch uniting all primates, with 30% favoring Dermoptera as sister to the Primate suborder Strepsirrhini and 31% placing Dermoptera sister to the Primate suborder Haplorrhini. Similarly, only a small plurality of genes and sites have histories that place Dermoptera as sister to the Primates rather than either of the 2 alternative topologies (gCF = 37, sCF = 4 0; Fig 1), despite the maximal statistical support for these relationships. While discordance at individual gene trees can result from technical problems in tree inference (e.g., long-branch attraction, low phylogenetic signal, poorly aligned sequences, or model misspecification), it also often reflects biological causes of discordance such as ILS and introgression. We further address the possible role of technical errors in generating patterns of discordance in the section entitled "Sources of gene tree discordance" below.

Within the Primates, the phylogenetic affiliation of tarsiers (represented here by *C. syrichta*) has been debated since the first attempts by Buffon (1765) and Linnaeus (1767 to 1770) to systematically organize described species [58]. Two prevailing hypotheses group tarsiers (Tarsiiformes) with either lemurs and lorises (the "prosimian" hypothesis [59]) or with Simiiformes (the "Haplorrhini" hypothesis [60], where Simiiformes = Apes+OWM+NWM). The ML-ASTRAL and ML-CONCAT analyses place Tarsiiformes with Simiiformes, supporting the Haplorrhini hypothesis (Fig 1). The Strepsirrhines come out as a well-supported group sister to the other primates. Again, our inference of species relationships is consistent with previous genomic analyses [61,62] but also highlights the high degree of discordance in this part of the tree. The rapid radiation of mammalian lineages that occurred in the late Paleocene and early Eocene [32] encompassed many of the basal primate branches, including the lineage leading to Haplorrhini. The complexity of this radiation is likely the reason for low gCF and sCFs (39.5% and 36%, respectively) for the branch leading to Haplorrhini and perhaps explains why previous studies recovered conflicting resolutions for the placement of tarsiers [31,63,64].

The remaining branches of the species tree that define major primate clades all have remarkably high concordance with the underlying gene trees (gCF >80%), though individual branches within these clades do not. The gCFs for the branches defining these clades are Strepsirrhini (lemurs+lorises) = 84.5, Catarrhini (OWM+Apes) = 90.0, Platyrrhini (NWM) = 96.6, Hominoidea (Apes) = 82.7, and Cercopithecidae (OWM) = 92.3 (Fig 1). High gene tree/species tree concordance for these branches is likely due to a combination of more recent divergences (increasing gene tree accuracy) and longer times between branching events [65].

Within these clades, however, we see multiple recent radiations. One of the most contentious has been among the NWMs, a set of relationships we address next.

## ML concatenation affects resolution of the New World monkey radiation

Sometime during the mid to late Eocene (approximately 45 to 34 mya), a small number of primates arrived on the shores of South America [15,66]. These monkeys likely migrated from Africa [66] and on arrival underwent multiple rounds of extinction and diversification [15]. Three extant families from this radiation now make up the NWMs (Platyrrhini; Fig 1). Because of the rapidity with which these species spread and diversified across the new continent, relationships at the base of the NWM have been hard to determine [12–14,16–18].

As reported above, the concatenated analysis (ML-CONCAT) gives a different topology than the gene tree-based analysis (ML-ASTRAL). Specifically, the ML-CONCAT analysis supports a symmetrical tree, with *Aotus* sister to *Callithrix* (Fig 2A). In contrast, ML-ASTRAL supports an asymmetrical (or "caterpillar") tree, with *Aotus* sister to a clade comprised of *Saimiri*+*Cebus* (Fig 2B). There are reasons to have doubts about both topologies. It is well known that carrying out maximum likelihood analyses of concatenated datasets can result in incorrect species trees, especially when the time between speciation events is short [67,68]. In fact, the specific error that is made in these cases is for ML concatenation methods to prefer a

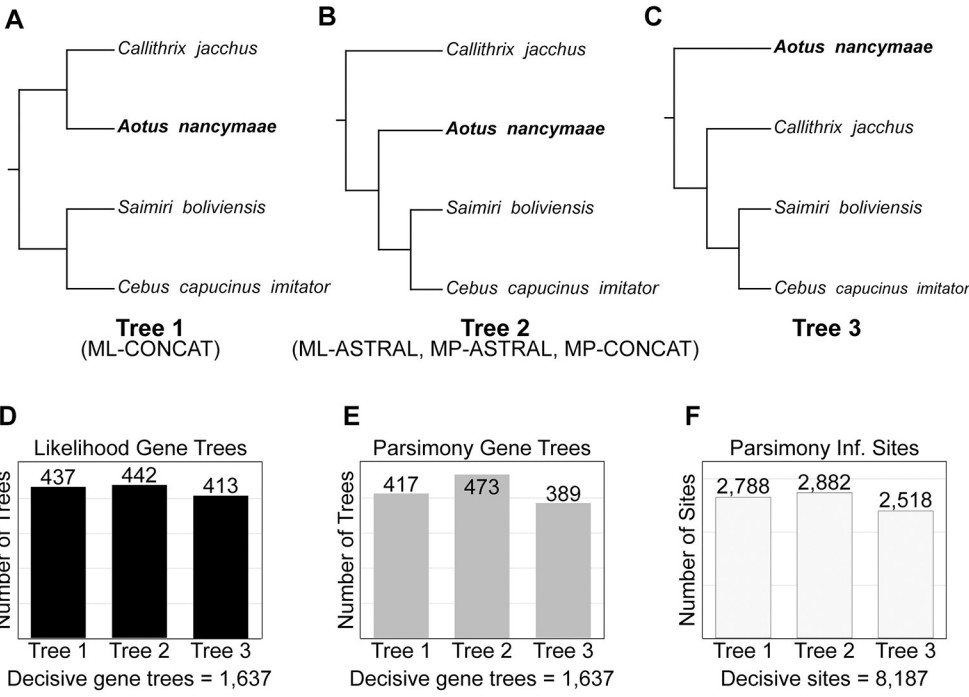

**Fig 2. The 3 most frequent topologies of NWMs.** (A) Tree 1 is the symmetrical topology inferred by the ML-CONCAT analysis of 1,730 loci (1.76 Mb). (B) Tree 2 is the asymmetrical topology inferred by ASTRAL III using either maximum likelihood (ML-ASTRAL) or maximum parsimony (MP-ASTRAL) gene tree topologies. Using maximum parsimony on the concatenated alignment also returns this tree (MP-CONCAT). (C) Tree 3 is the alternative resolution recovered at high frequency in all gene tree analyses, though it is not the optimal species tree using any of the methods. (D) Number of gene trees supporting each of the 3 resolutions of the NWM clade when maximum likelihood is used to infer gene tree topologies. There are 1,637 decisive gene trees for these splits. (E) Gene tree counts when maximum parsimony is used to infer gene tree topologies. (F) Number of parsimony informative sites in the concatenated alignment supporting each of the 3 resolutions. ML-CONCAT, maximum likelihood concatenated; NWM, New World monkey.

symmetrical 4-taxon tree over an asymmetrical one, exactly as is observed here. Gene tree-based methods such as ASTRAL are not prone to this particular error, as long as the underlying gene trees are all themselves accurate [69,70]. However, if there is bias in gene tree reconstruction, then there are no guarantees as to the accuracy of the species tree. In addition, the ML-ASTRAL tree is supported by only a very small plurality of gene trees: There are 442 trees supporting this topology, compared to 437 supporting the ML-CONCAT topology and 413 supporting the third topology (Fig 2D). This small excess of supporting gene trees also explains the very low posterior support for this branch in the species tree (Fig 1). Additionally, a polytomy test [71], implemented in ASTRAL and performed using ML gene trees, failed to reject the null hypothesis of "polytomy" for the branch uniting *Aotus*+(*Saimiri*,*Cebus*) ($P = 0.47$).

To investigate these relationships further, we carried out additional analyses. The trees produced from concatenated alignments can be biased in situations with high ILS when maximum likelihood is used for inference, but this bias does not affect parsimony methods [21,72]. Therefore, we analyzed exactly the same concatenated 1.76 Mb alignment used as input for ML but carried out a maximum parsimony analysis in PAUP* [73]. As would be expected given the known biases of ML methods, the maximum parsimony tree (which we refer to as "MP-CONCAT") returns the same tree as ML-ASTRAL, supporting an asymmetric topology of NWMs (Fig 2B). Underlying this result is a relatively large excess of parsimony informative sites supporting this tree (Fig 2F), which results in maximal bootstrap values for every branch. The 2 most diverged species in this clade (*Saimiri* and *Callithrix*) are only 3.26% different at the nucleotide level, so there should be little effect of multiple substitutions on the parsimony analysis.

As mentioned above, gene tree-based methods (such as ASTRAL) are not biased when accurate gene trees are used as input. However, in our initial analyses, we used maximum likelihood to infer the individual gene trees. Because protein-coding genes are themselves often a combination of multiple different underlying topologies [74], ML gene trees may be biased and using them as input to gene tree-based methods may still lead to incorrect inferences of the species tree [75]. Therefore, we used the same 1,730 loci as above to infer gene trees using maximum parsimony with MPBoot [76]. Although the resulting topologies still possibly represent the average over multiple topologies contained within a protein-coding gene, using parsimony ensures that this average tree is not a biased topology. These gene trees were used as input to estimate a species tree using ASTRAL; we refer to this as the "MP-ASTRAL" tree. Once again, the methods that avoid known biases of ML lend further support to an asymmetric tree, placing *Aotus* sister to the *Saimiri*+*Cebus* clade (Fig 2B). In fact, the gene trees inferred with parsimony now show a much greater preference for this topology, with a clear plurality of gene trees supporting the species tree (473 versus 417 supporting the second most common tree; Fig 2E). As a consequence, the local posterior for this branch in the MP-ASTRAL tree is 0.92, and the polytomy test performed using MP gene trees rejects ($P = 0.037$) the null hypothesis of "polytomy" for the branch uniting *Aotus*+(*Saimiri*,*Cebus*). The increased number of concordant gene trees using parsimony suggests that the gene trees inferred using ML may well have been suffering from the biases of concatenation when multiple trees are brought together (as observed in the Great Apes [74]), reducing the observed levels of concordance.

A recent analysis of NWM genomes found *Aotus* sister to *Callithrix*, as in the ML-CONCAT tree, despite the use of gene trees to build the species tree [18]. However, the outgroup used in this analysis is a closely related species (*Brachyteles arachnoides*) that diverged during the NWM radiation and that shares a recent common ancestor with the ingroup taxa [12,13]. If the outgroup taxon used to root a tree shares a more recent common ancestor with subsets of ingroup taxa at an appreciable number of loci, the resulting tree topologies will be biased. A similar problem likely arose in previous studies that have used the Scandentia or Dermoptera

as outgroups to Primates. In general, this issue highlights the difficulty in choosing outgroups: Though we may have 100% confidence that a lineage lies outside our group of interest in the species tree, a reliable outgroup must also not have any discordant gene trees that place it inside the ingroup.

## Sources of gene tree discordance

As previously mentioned, there are both biological reasons for gene tree discordance (e.g., ILS or introgression) and technical reasons (e.g., long-branch attraction, homoplasy, low phylogenetic signal, poorly aligned sequences, or model misspecification). All of these phenomena may be reflected in gCFs and sCFs, but the proportion of discordance attributable to biological versus technical factors is often difficult to ascertain. We therefore performed additional analyses to assess the impact of error on estimates of concordance factors.

In order to determine the degree to which short alignments or genes with low phylogenetic signal contribute to inaccurate gene trees, we recalculated gCFs and sCFs using the genes with the 200 longest alignments in our dataset (lengths ranging from 1,640 bp to 6,676 bp, with 116 to 2,101 parsimony informative sites). The resulting gCFs for the branch leading to the Primate common ancestor increases from 45% to 66%, while the sCFs remain unchanged (S1 Fig). For the branch placing Dermoptera sister to Primates, using trees estimated from the 200 longest alignments resulted in a modest increase in gCFs from 37% to 45%. Overall, the gCFs for the 200 longest genes were higher for all branches in the tree, with the average gCF increasing from 65.18% to 79.74%. The consistent increase in gCF but not sCF when using longer genes points to errors in gene tree inference as a small, but significant, factor in our dataset.

Using a single outgroup (mouse) could potentially lead to biases such as long-branch attraction near the base of the tree. To ameliorate these concerns, we performed an additional analysis using 150 randomly chosen single-copy orthologs, with pika (*Ochotona princeps*) included as a second outgroup. As in the full dataset, maximum likelihood and parsimony were both applied to a concatenated dataset, and gene trees were also inferred via both ML and parsimony. Parsimony analysis of the concatenated alignment resulted in the same topology as in Fig 1, while a maximum likelihood analysis produced the same topology as the full ML-CON-CAT tree from 1,730 loci, preferring a symmetric tree for the NWM clade. To assess the effect of including an additional outgroup on concordance factors, we calculated gCFs and sCFs using the 150 single-copy orthologs both with (S2A Fig) and without (S2B Fig) pika (using ML gene trees). In contrast to expectations about any error introduced by long-branch attraction, we observe slightly lower gCFs near the base of the tree when pika is included (S2A Fig). sCFs are not affected by the inclusion of pika. These analyses indicate that including additional outgroups when analyzing the full dataset is unlikely to reduce concordance factors or to change inferences of the species tree.

Technical errors leading to discordance should be more prominent deeper in the tree, as there is more opportunity for long-branch attraction, homoplasy, poor alignments, or model misspecification to cause problems. To determine whether concordance factors for deep branches in the primate tree are disproportionately affected by error, we looked for a correlation between concordance factors and the age of each bifurcation in the tree. For gCFs, we found no correlation with node age ($r^2 = 0.0094$), while sCFs were slightly negatively correlated ($r^2 = 0.2998$; S3 Fig). The negative correlation found between sCFs and node age is consistent with the expectation that substitutions occurring on deeper branches of the tree are more likely to suffer from the effects of multiple substitutions (homoplasy). While there may still be technical factors affecting gCFs, true discordance throughout the tree is high enough to mask any such effect.

A recent simulation study [77] reported that negative selection, in combination with large differences in effective population size, can generate strong enough asymmetries in gene tree topologies that the most common topology does not match the species tree. Such an effect, if real, would mislead both gene tree-based and concatenation-based approaches to species tree inference. However, previous theoretical results predict that there should be no effect of negative selection on the distribution of tree topologies [78–81], and the new results were obtained using custom simulation software. To clarify this issue, we used the open-source simulator SLiM [82] to study non-recombining loci under the most extreme parameters used by He and colleagues [77] (see Materials and methods). We found no evidence for the bias in gene tree frequencies recently reported (S4A Fig). However, we observed fewer than 1 mutation per locus at the end of our simulations under the parameters exactly replicating He and colleagues [77], suggesting we may not have generated sufficient deleterious variation to observe the effect. To address this, we simulated the same conditions but with the deleterious mutation rate increased by 2 orders of magnitude and still did not observe a bias in topology frequencies (S4B Fig). Our results therefore indicate that weak negative selection does not generate gene tree discordance, consistent with population genetic theory [78–81].

## Strongly supported divergence times using fossil calibrations

Fossil-constrained molecular dating was performed using 10 independent datasets, each of which consisted of 40 protein-coding genes randomly selected (without replacement) and concatenated. The resulting datasets had an average alignment length of 39,374 bp (SD = $2.6 \times 10^3$; S5 Table). Although individual discordant trees included in this analysis may have different divergence times, the difference in estimates of dates should be quite small [83]. We used 8 dated fossils (blue stars in Fig 1) from 10 studies for calibration (S6 Table). The most recent of these fossils is approximately 5.7 mya [84], while the most ancient is 55.8 mya [85]. Each separate dataset and the same set of "soft" fossil constraints, along with the species tree in Fig 1, were used as input to PhyloBayes 3.3 [86], which was run twice to assess convergence (Materials and methods).

We observed tight clustering of all estimated node ages across datasets and independent runs of PhyloBayes (Fig 3 and S6 Table). In addition, the ages of most major crown nodes estimated here are largely in agreement with previously published age estimates (Table 2). Some exceptions include the age of the crown Strepsirrhini (47.4 mya) and Haplorrhini (59.0 mya), which are more recent than many previous estimates for these nodes (range in the literature is Strepsirrhini = 51.6 to 68.7, Haplorrhini = 60.6 to 81.3; see Table 2). The crown nodes for Catarrhini, Hominoidea, and Cercopithecidae (28.4, 21.4, and 16.8 mya, respectively) all fall within the range of variation recovered in previous studies (Table 2).

Our estimate for the most recent common ancestor of the extant primates (i.e., the last common ancestor of Haplorrhini and Strepsirrhini) is 61.7 mya, which is slightly more recent than several studies [13,31,33,88] and much more recent than other studies [12,87,89] (Table 2). However, our estimate is in good agreement with Herrera and colleagues [32], who used 34 fossils representing extinct and extant lineages (primarily Strepsirrhines) to infer divergence times among primates, concluding that the split occurred approximately 64 mya. Despite limited overlap in taxon sampling, 1 similarity between our study and that of Herrera and colleagues is that we have both used the maximum constraint of 65.8 million years (my) on the ancestral primate node suggested by Benton and colleagues [90], which likely contributes to the more recent divergence. It is worth noting that the soft bounds imposed in our analysis permit older ages to be sampled from the Markov chain, but these represented only a small fraction (median 3.37%) of the total sampled states after burn-in (S6 Table). To

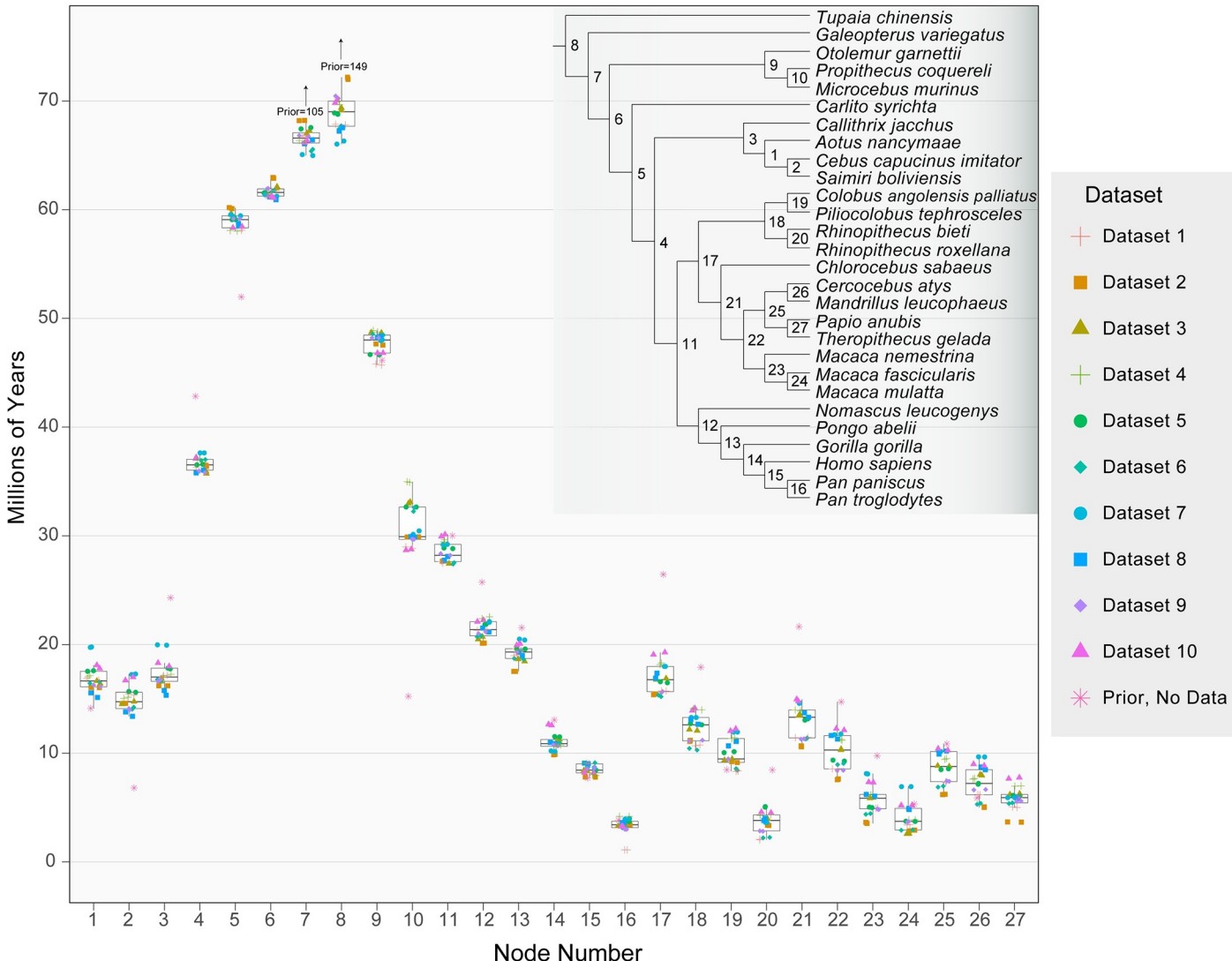

**Fig 3. Mean node ages for independent Phylobayes dating runs.** Box plots show the median, interquartile range, and both minimum and maximum values of the mean nodes ages for 10 different datasets (with each dataset run twice). An additional run was performed with no sequence data to ascertain the prior on node divergence times in the presence of fossil calibrations (pink asterisks). Some prior ages were too large to include in the plot while still maintaining detail; these ages are given as numeric values. The species tree topology is from Fig 1; 95% HPD intervals for each node are reported in S7 Table. Node age estimates for each independent PhyloBayes run are provided in S1 Data. HPD, highest posterior density.

determine the effects of imposing the 65.8 my maximum constraint on the Primate node, we analyzed all 10 datasets for a third time with this constraint removed and report the divergence time of major primate clades in Table 2 ("No Max" entries). However, it may be that using genes that have gene trees most similar to the topology being dated will reduce bias caused by concatenation [74]. To determine whether using concordant loci has an impact on the estimated dates, we constructed an 11th dataset consisting of approximately 43 kb from the 20 loci most similar to the species tree in Fig 1 (as determined by Robinson–Foulds distances). There was no consistent difference in the dates estimated with this dataset ("Concord" entries in Table 2).

There are several caveats to our age estimates that should be mentioned. Maximum age estimates for the crown node of any given clade are defined by the oldest divergence among

**Table 2. Mean crown node divergence times estimated in this study compared with mean divergences times estimated by 8 prior studies.**

| Node | This study | This study, no max[*] | This study, concord[†] | Herrera et al. [32] | Kistler et al. [33] | Perez et al. [17] | Springer et al. [13] | Meredith et al. [45] | Perelman et al. [12] | Wilkinson et al. [87] | Chatterjee et al. [31] |
|---|---|---|---|---|---|---|---|---|---|---|---|
| Primates | 61.7 | 67.5 | 63 | 63.9 | 68 | NA | 67.8 | 71.5 | 87.2 | 84.5 | 63.7 |
| Strepsirrhini | 47.4 | 50.2 | 48.4 | 61.4 | 59 | NA | 54.2 | 55.1 | 68.7 | 49.8 | 51.6 |
| Haplorrhini | 59.0 | 63.8 | 59.8 | 61.9 | 67 | 60.6 | 61.2 | 62.4 | 81.3 | NA | NA |
| Catarrhini | 28.4 | 29.0 | 27.2 | 32.1 | 33 | 27.8 | 25.1 | 20.6 | 31.6 | 31.0 | 29.3 |
| Hominoidea | 21.4 | 21.6 | 19.9 | NA | 21 | 18.44 | 17.4 | 14.4 | 20.3 | NA | 21.5 |
| Cercopithecidae | 16.8 | 16.9 | 14.2 | NA | 24 | 13.4 | 13.2 | NA | 17.6 | 14.1 | 23.4 |

Estimates were calculated by averaging the mean times across all runs for 10 independent datasets.

[*]Refers to the average divergence time of the crown node for the indicated taxonomic group when the 65.8 my maximum constraint was removed from the Primate node.

[†]Refers to the average divergence time of the crown node for the indicated taxonomic group when divergence times were estimated using the most concordant gene trees. Datasets used in all dating analyses are available via Data Dryad in the archive All_Dating_Datasets_DRYAD.tar.gz, https://doi.org/10.5061/dryad.rfj6q577d [22]. my, million years; NA, not applicable.

sampled taxa in the clade. This limitation results in underestimates for nearly all crown node ages as, in practice, complete taxon sampling is difficult to achieve. Fossil calibrations are often employed as minimum constraints in order to overcome the limitations imposed by taxon sampling, allowing older dates to be estimated more easily. On the other hand, the systematic underestimation of crown node ages due to taxon sampling is somewhat counteracted by the overestimation of speciation times due to ancestral polymorphism. Divergence times estimated from sequence data represent the coalescence times of sequences, which are necessarily older than the time at which 2 incipient lineages diverged [91,92]. This overestimation will have a proportionally larger effect on recent nodes (such as the *Homo/Pan* split; Fig 3, node 15), but the magnitude can be no larger than the average level of polymorphism in ancestral populations and will be additionally reduced by post-divergence gene flow.

## Introgression during the radiation of primates

There is now evidence for recent interspecific gene flow between many extant primates, including introgression events involving humans [25], gibbons [93,94], baboons [9,27], macaques [95,96], and vervet monkeys [10], among others. While there are several widely used methods for detecting introgression between closely related species (see chapters 5 and 9 in [97]), detecting ancient gene flow is more difficult. One of the most popular methods for detecting recent introgression is the *D* test (also known as the "ABBA-BABA" test; [98]). This test is based on the expectation that, for any given branch in a species tree, the 2 most frequent alternative resolutions should be present in equal proportions. However, the *D* test uses individual SNPs to evaluate support for alternative topologies and explicitly assumes an infinite sites model of mutation (i.e., no multiple hits). As this assumption will obviously not hold the further back in time one goes, a different approach is needed.

Fortunately, Huson and colleagues [36] described a method that uses gene trees themselves (rather than SNPs) to detect introgression. Using the same expectations as in the *D* test, these authors looked for a deviation from the expected equal numbers of alternative tree topologies using a test statistic they refer to as Δ. As far as we are aware, Δ has only rarely been used to test for introgression in empirical data, possibly because of the large number of gene trees needed to assess significance or the assumptions of the parametric method proposed to obtain *P*

values. Here, given our large number of gene trees and large number of internal branches to be tested, we adapt the Δ test for genome-scale data.

To investigate patterns of introgression within primates, we used 1,730 single-copy loci to test for deviations from the null expectation of Δ on each of the 24 internal branches of the primate phylogeny (Materials and methods). To test whether deviations in Δ were significant (i.e., $\Delta > 0$), we generated 2,000 resampled datasets of 1,730 gene tree topologies each. *P* values were calculated from *Z*-scores generated from these resampled datasets. Among the 17 branches where at least 5% of topologies were discordant, we found 7 for which Δ had $P < 0.05$.

To further verify these instances of potential introgression, for each of these 7 branches we increased the number of gene trees used, as well as the alignment length for each locus, by sub-sampling a smaller set of taxa. We randomly chose 4 taxa for each internal branch tested that also had this branch as an internal branch and then aligned all orthologs present in a single copy in each taxon. These steps resulted in approximately 3,600 to 6,400 genes depending on the branch being tested (S8 Table). Additionally, because instances of hybridization and introgression are well documented among macaques [96,99,100], we similarly resampled orthologs from the 3 *Macaca* species in our study.

We recalculated Δ using the larger gene sets and found significant evidence (after correcting for *m* = 17 multiple comparisons by using a cutoff of *P* = 0.00301) for 6 introgression events, all of which occurred among the Papionini (Fig 4 and see next paragraph). Within the Hominoidea, we found Δ = 0.0518 for the branch leading to the great apes (*P* = 0.030). The asymmetry in gene tree topologies here suggests that gene flow may have happened between gibbons (represented by *Nomascus*) and the ancestral branch leading to the African hominoids

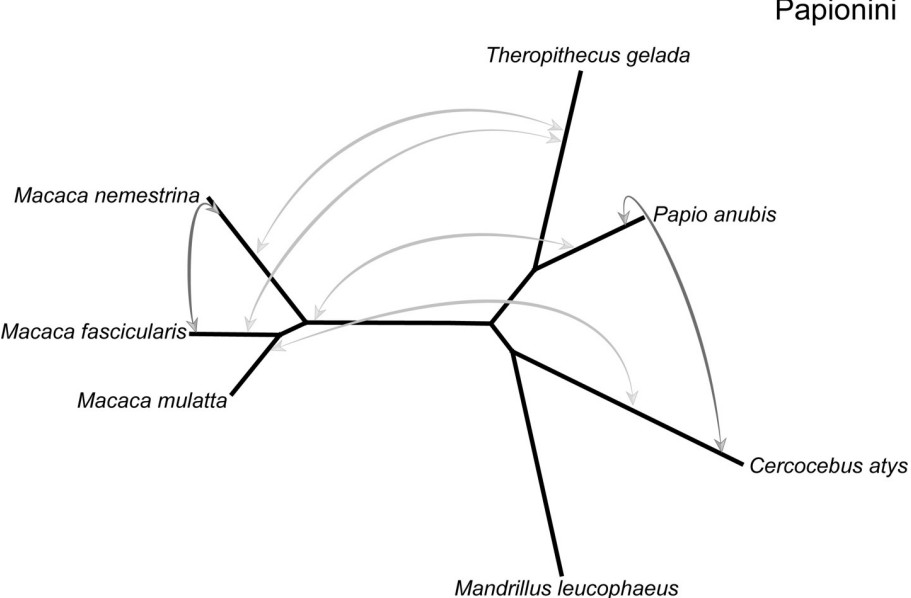

**Fig 4. Introgression among Papionini taxa (the species tree is unrooted for clarity).** Arrows indicate that a significant Δ was found in our 4 taxon tests and identify the 2 lineages inferred to have exchanged genes (values underlying these tests are listed in S8 Table). Among the Papionini, there was evidence of introgression between African taxa (*Papio*, *Theropithecus*, and *Cercocebus*) and Asian *Macaca* species (light gray arrows). Introgression events likely occurred between African taxa and the ancestral *Macaca*, which had a wide distribution across Northern Africa prior to the radiation throughout Asia 2–3 mya [103]. More recent instances of introgression are inferred between macaque species and among the African Papionini (dark gray arrows). mya, million years ago.

(humans, chimpanzees, and gorillas), but, like the *D* test, Δ cannot tell us the direction of introgression. Although currently separated by significant geographic distances (African apes south of the Sahara Desert and gibbons all in Southeast Asia), it is worth noting that fossil hominoids dating from the early to late Miocene had a broad distribution extending from Southern Africa to Europe and Asia [101]. Support for introgression between ancestral hominins and ancestral chimpanzees has been previously reported [102]; our 4-taxon analyses found marginal support for this conclusion (Δ = 0.0917, *P* = 0.055).

Within the OWM, approximately 40% of Cercopithicine species are known to hybridize in nature [34]. Consistent with this, *M. nemestrina* and *Macaca fascicularis* showed a strong signature of gene flow in our data (Δ = 0.1761, *P* = 1.377e-09). These 2 species have ranges that currently overlap (S5 Fig). In contrast to the clear signal of recent gene flow in the macaques, we detected a complex pattern of ancient introgression between the African Papionini (*Cercocebus*, *Mandrillus*, *Papio*, *and Theropithecus*) and the Asian Papionini (*Macaca*) (Fig 4). The Δ test was significant using multiple different subsamples of 4 taxa, suggesting multiple ancestral introgression events. An initial attempt to disentangle these events using Phylonet v3.8.0 [104] with the 7 Papionini species and an outgroup was unsuccessful, as Phylonet failed to converge on an optimal network for these taxa. An attempt to infer the network with SNaQ [105] gave similarly ambiguous results. When there are multiple episodes of gene flow within a clade, even complex computational machinery may be unable to infer the correct combination of events.

As an alternative approach, we used 4-taxon trees to estimate Δ for each *Macaca* species paired with 2 African Papionini (1 from the *Papio+Theropithecus* clade and 1 from the *Mandrillus* +*Cercocebus* clade; see S8 Table) and an outgroup. Significant introgression was detected using each of the *Macaca* species and 3 of the 4 African Papionini species (*Cercocebus*, *Theropithecus*, and *Papio*). These results suggest gene flow between the ancestor of the 3 *Macaca* species in our analysis and the ancestors of the 3 African Papionini in our analysis, or 1 introgression event involving the ancestor of all 4 African species coupled with a second event that masked this signal in *Mandrillus*. This second event may have been either biological (additional introgression events masking the signal) or technical (possibly the lack of continuity or completeness of the *Mandrillus* reference genome sequence), but in either case, we could not detect introgression in the available drill sequence. The latter scenario would fit better with the current geographic distributions of these species, as they are on 2 different continents. However, the fossil record indicates that by the late Miocene to late Pleistocene, the ancestral distribution of the genus *Macaca* covered all of North Africa, into the Levant, and as far north as the United Kingdom (S5 Fig; [106]). The fossil record for *Theropithecus* indicates that several species had distributions that overlapped with *Macaca* during this time, including in Europe and as far east as India (S5 Fig; [107,108]). Ancestral macaques and ancestral Papionini may therefore have come into contact in the area of the Mediterranean Sea. The Sahara Desert is also responsible for the current disjunct distributions of many of these species. However, this region has experienced periods of increased rainfall or "greenings" over the past several million years [109–111]. Faunal migration through the Sahara, including by hominins, is hypothesized to have occurred during these green periods [110,112,113], resulting in successive cycles of range expansion and contraction [114]. Hybridization and introgression could have occurred between the ancestors of these groups during 1 of these periods.

Our results on introgression come with multiple caveats, both about the events we detected and the events we did not detect. As with the *D* test, there are multiple alternative explanations for a significant value of Δ besides introgression. Ancestral population structure can lead to an asymmetry in gene tree topologies [115] though it requires a highly specific, possibly unlikely population structure. For instance, if the ancestral population leading to *M. nemestrina* was

more closely related to *M. fascicularis* than was the ancestral population leading to its sister species, *M. mulatta* (Fig 4), then there could be an unequal number of alternative topologies. Similarly, any bias in gene tree reconstruction that favors 1 alternative topology over the other could potentially lead to a significant value of Δ. While this scenario is unlikely to affect recent divergences using SNPs, well-known biases that affect topology reconstruction deeper in the tree (such as long-branch attraction) could lead to gene tree asymmetries. However, we did not observe any significant Δ-values for branches more than approximately 10 my old. One alternative approach to avoid biases in reconstruction could be the use of transposon insertions or other rare genomic changes (cf. [116,117]). Future analyses that compare these different approaches to detecting introgression would be especially useful.

There are also multiple reasons why our approach may have missed introgression events, especially deeper in the tree. All methods that use asymmetries in gene tree topologies miss gene flow between sister lineages, as such events do not lead to changes in the proportions of underlying topologies. Similarly, equal levels of gene flow between 2 pairs of non-sister lineages can mask both events, while even unequal levels will lead one to miss the less frequent exchange. More insidiously, especially for events further back in time, extinction of the descendants of hybridizing lineages will make it harder to detect introgression (though extinction of donor lineages is less of a problem than extinction of lineages receiving migrants). Internal branches closer to the root will be on average longer than those near the tips because of extinction [118], and therefore, introgression between non-sister lineages would have to occur longer after speciation in order to be detected. For instance, gene flow among Strepsirrhine species has been detected in many previous analyses of more closely related species (e.g., [119–122]), but the deeper relationships among the taxa sampled here may have made it very difficult to detect introgression. Nevertheless, our analyses were able to detect introgression between many primate species across the phylogeny.

## Conclusions

Several previous phylogenetic studies of primates have included hundreds of taxa, but fewer than 70 loci [12,13]. While the species tree topologies produced by these studies are nearly identical to the one recovered in our analysis, the limited number of loci meant that it was difficult to assess gene tree discordance accurately. By estimating gene trees from 1,730 single-copy loci, we were able to assess the levels of discordance present at each branch in the primate phylogeny. Understanding discordance helps to explain why there have been long-standing ambiguities about species relationships near the base of primates and in the radiation of NWMs. Our analyses reveal how concatenation of genes—or even of exons—can mislead maximum likelihood phylogenetic inference in the presence of discordance, but also how to overcome these biases. Discordance also provides a window into introgression among lineages, and here, we have found evidence for exchange among several species pairs. Each instance of introgression inferred from the genealogical data is plausible insofar as it can be reconciled with current and ancestral species distributions.

## Materials and methods

### Source material and sequencing

The San Diego Zoo and the Washington National Primate Research Center provided biomaterials to the Human Genome Sequencing Center (HGSC), Baylor College of Medicine under an agreement that granted permission to the HGSC to use the biomaterials for academic scientific research. This is the standard agreement between these institutions which regularly provide this service and the academic community that uses their biomaterials for various types of

analyses. The HGSC does not pay for biomaterials but does cover the costs of shipping the bio-materials from the provider Baylor.

For the sequencing of the *C. angolensis palliatus* genome, paired-end (100 bp) libraries were prepared using DNA extracted from heart tissue (isolate OR3802 from the San Diego Zoo). Sequencing was performed using 9 Illumina Hi-seq 2000 lanes and 4 Illumina Hi-seq 2500 lanes with subsequent assembly carried out using ALLPATHS-LG software (v. 48744) [123]. Additional scaffolding and gap-filling was performed using Atlas-Link v. 1.1 (https://www.hgsc.bcm.edu/software/atlas-link) and Atlas-GapFill v. 2.2. (https://www.hgsc.bcm.edu/software/atlas-gapfill), respectively. Annotation for all 3 species was carried out using the NCBI Eukaryotic Genome Annotation Pipeline. A complete description of the pipeline can be viewed at https://www.ncbi.nlm.nih.gov/genome/annotation_euk/process/.

For the sequencing of the *M. nemestrina* genome, DNA was extracted from a blood sample (isolate M95218 from the Washington National Primate Research Center). Paired-end librar-ies were prepared and sequenced on 20 Illumina Hi-Seq 2000 lanes with the initial assembly performed using ALLPATHS-LG as above. Scaffolding was conducted using Atlas-Link v. 1.1. Additional gap-filling was performed using the original Illumina reads and Atlas-GapFill v. 2.2, as well as long reads generated using the Pacific Biosciences RS (60 SMRT cells) and RSII (50 SMRT cells) platforms. The PacBio reads were mapped to scaffolds to fill remaining gaps in the assembly using PBJelly2 (v. 14.9.9) [124].

For the sequencing of the *M. leucophaeus* genome, DNA was extracted from heart tissue (isolate KB7577 from the San Diego Zoo). Paired-end libraries were prepared and sequenced on 9 Illumina Hi-Seq 2000 lanes with the initial assembly performed using ALLPATHS-LG as above. Additional scaffolding was completed using Atlas-Link v. 1.1, and additional gap-filling in scaffolds was performed using the original Illumina reads and Atlas-GapFill v. 2.2.

## Phylogenomic analyses

The full set of protein-coding genes for 26 primates and 3 non-primates were obtained by combining our newly sequenced genomes with already published data (see S1 Table for refer-ences and accessions and Table 1 and S2 Table for genome statistics). Ortholog clustering was performed by first executing an all-by-all BLASTP search [125,126] using the longest isoform of each protein-coding gene from each species. The resulting BLASTP output was clustered using the mcl algorithm [127] as implemented in FastOrtho [128] with various inflation parameters (the maximum number of clusters was obtained with *inflation = 5*). Orthogroups were then parsed to retain those genes present as a single copy in all 29 taxa (1,180 genes), 28 of 29 taxa (1,558 genes), and 27 of 29 taxa (1,735 genes). We chose to allow up to 2 missing spe-cies per alignment to maximize the data used in our phylogenomic reconstructions while maintaining high taxon occupancy in each alignment.

CDS for each single-copy orthogroup were aligned, cleaned, and trimmed via a multistep pro-cess: First, sequences in each orthogroup were aligned by codon using GUIDANCE2 [129] in con-junction with MAFFT v7.407 [130] with 60 bootstrap replicates. GUIDANCE2 uses multiple bootstrapped alignments to generate quality scores for each column in the final alignment as well as for each taxon sequence in each alignment. Sequence residues in the resulting MAFFT align-ment with GUIDANCE scores <0.93 were converted to gaps, and sites with >50% gaps were removed using Trimal v1.4.rev22 [131]. Alignments shorter than 200 bp (full dataset) or 300 bp (4-taxon tests for introgression), and alignments that were invariant or contained no parsimony informative characters, were removed from further analyses. Alignments with high numbers of discordant sites were further inspected for errors and removed from the analysis when warranted. This resulted in 1,730 loci for the full analysis (see S8 Table for gene counts used in 4-taxon tests).

IQ-TREE v2-rc1 was used with all 1,730 aligned loci to estimate a maximum likelihood concatenated tree with an edge-linked, proportional-partition model, and 1,000 ultrafast bootstrap replicates [132,133]. This strategy uses ModelFinder [134] to automatically find the best-fit model for each ortholog alignment (partition). Branch lengths are shared between partitions, with each partition having its own rate that rescales branch lengths, accommodating different evolutionary rates between partitions. The full IQ-TREE command line used was "iqtree -p Directory_of_Gene_Alignments--prefix -m MFP -c 8 -B 1000". Maximum likelihood gene trees were estimated for each alignment with nucleotide substitution models selected using ModelFinder [134] as implemented in IQ-TREE. The full IQ-TREE command line used was "iqtree -s Directory_of_Gene_Alignments--prefix -m MFP -c 8". We used the resulting maximum likelihood gene trees to estimate a species tree using ASTRAL III (ML-ASTRAL) [39]. Parsimony gene trees were generated using MPboot [76] and used to estimate a species tree using ASTRAL III (MP-ASTRAL), while PAUP* [73] was used to estimate the concatenated parsimony tree (MP-CONCAT) with 500 bootstrap replicates. IQ-TREE was used to calculate both gCFs and sCFs, with sCFs estimated from 300 randomly sampled quartets using the command line "iqtree--cf-verbose--gcf 1730_GENETREE.treefile -t Species_tree_file--df-tree--scf 300 -p Directory_of_Gene_Alignments -c 4".

### Effects of selection on gene tree distributions

We performed 100 replicate simulations for each mutation rate condition using SLiM version 3.3.1 [82], with tree sequence recording turned on and no neutral mutations. Each replicate simulation consisted of 50 non-recombining loci of 1 kb each, with free recombination between loci, for 3 populations with the phylogenetic relationship ((p2,p3),p1). These simulations closely match the population genetic parameters under the most extreme asymmetry condition reported in He and colleagues [77], with population sizes, selection coefficients, and mutation rates rescaled 2 orders of magnitude for performance (SLiM recipes are available via Data Dryad: https://doi.org/10.5061/dryad.rfj6q577d [22]). These parameters include a per-locus deleterious mutation rate of $3 \times 10^{-7}$ per generation; a population-scaled selection coefficient ($Ns$) of $-7.5$; an internal branch subtending p2 and p3 of $0.01N$ generations (where $N$ is the population size of p1 and p2); a population size for p3 that is 0.04 times than that of p1 and p2; and tip branch lengths of $8N$ generations. In the higher mutation rate condition, the per-locus rate was increased to $3 \times 10^{-5}$ per generation. We randomly sampled 1 chromosome from each population at each locus at the end of the simulation and obtained the genealogy of these samples recorded in the tree sequence at the locus.

### Introgression analyses

For each internal branch of the primate tree where the proportion of discordant trees was >5% of the total, concordance factors were used to calculate the test statistic Δ, where

$$\Delta = \frac{Number\ of\ DF1\ trees - Number\ of\ DF2\ trees}{Number\ of\ DF1\ trees + Number\ of\ DF2\ trees},$$

where *DF1* trees represent the most frequent discordant topology, and *DF2* trees are the second most frequent discordant topology. This is a normalized version of the statistic proposed by Huson and colleagues [36], which only included the numerator of this expression. Note also that, by definition, Δ here is always equal to or greater than 0. To test whether deviations from zero were significant (i.e., Δ > 0), we calculated Δ for 2,000 pseudo-replicate datasets generated by resampling gene trees with replacement. The resulting distribution was used to calculate *Z*-scores and the resulting *P* values for the observed Δ value associated with each branch

tested [135]. Of the 17 internal branches where >5% of topologies were discordant, 7 were significant at $P < 0.05$, and selected for more extensive testing. For each of the 7 significant branches in the all-Primates tree, 4 taxa were selected that included the target branch as an internal branch. Single-copy genes present in each taxon were aligned as previously described. Alignments with no variant or parsimony informative sites were removed from the analysis, and gene trees were estimated using maximum likelihood in IQ-TREE 2. The test statistic, Δ, was calculated, and significance was again determined using 2,000 bootstrap replicates with the *P* value threshold for significance corrected for multiple comparisons ($m = 17$) using the Dunn–Šidák correction [136,137].

## Molecular dating

Molecular dating analyses were performed on 10 datasets consisting of 40 CDS alignments each sampled randomly without replacement from the 1,730 loci used to estimate the species tree. Gene alignments were concatenated into 10 supermatrices ranging from 36.7 kb to 42.7 kb in length (see S5 Table for the length of each alignment). Each dataset was then analyzed using PhyloBayes 3.3 [86] with sequences modeled using a site-specific substitution process with global exchange rates estimated from the data (CAT-GTR; [138]). Among-site rate variation was modeled using a discrete gamma distribution with 6 rate categories. A relaxed molecular clock [139] with 8, soft-bounded, fossil calibrations (see S6 Table) was used to estimate divergence times on the fixed species tree topology (Fig 1); the analyses were executed using the following command line: pb -x 1 15000 -d Alignment.phy -T Tree_file.tre -r outgroup_file. txt -cal 8_fossil.calib -sb -gtr -cat -bd -dgam 6 -ln -rp 90 90. Each dataset was analyzed for 15,000 generations, sampling every 10 generations, with 5,000 generations discarded as burn-in. Each dataset was analyzed twice to ensure convergence of the average age estimated for each node (Fig 3 shows the node age for both runs). To determine the effect of including a maximum constraint on the root of the Primates, we analyzed each dataset a third time with this constraint removed. Both the constrained and unconstrained node ages for major groups within the Primates are reported in Table 2.

Single-copy CDS gene alignments, gene trees, dating datasets, SLiM3 recipes, unaligned gene sequences, and PAUP commands can be accessed via the Data Dryad repository located at https://doi.org/10.5061/dryad.rfj6q577d [22].

## Supporting information

**S1 Fig. Concordance factors for the species tree in Fig 1 calculated using maximum likelihood gene trees and site patterns from the 200 longest single-copy loci alignments used in the 1,730-gene analysis.** In general, gCFs increase, while the sCFs remain the same, indicating that gene tree error is a likely source of some discordance. gCF, gene concordance factor; sCF, site concordance factor.
(PDF)

**S2 Fig. Concordance factors calculated using 150 randomly chosen single-copy orthologs, with pika (*Ochotona princeps*) included as an additional outgroup to mouse.** (A) gCFs and sCFs for these 150 genes when pika is included. (B) gCFs and sCFs for these same genes when pika is not included. We observe slightly higher gCFs near the base of the tree with pika excluded (red boxes). Note that these species trees use unit-length branch lengths for readability of branch labels. gCF, gene concordance factor; sCF, site concordance factor.
(PDF)

**S3 Fig. Gene and site concordance factors plotted as a function of node depth (in millions of years).** No correlation was found between gCFs and node depth, whereas a slightly negative correlation was found between sCFs and node depth. This relationship indicates that homoplasy may act to slightly reduce sCFs deeper in the tree. The data underlying mean node ages are provided in S1 Data. gCF, gene concordance factor; sCF, site concordance factor.
(PDF)

**S4 Fig. Forward simulations using SLiM3 with the most extreme parameters used by He et al.** (2020): population size combination "F" with $s = -7.5 \times 10^{-6}$ and $\Delta\tau = 2,000$. Our results show no significant difference in the distribution of gene tree topologies in the presence of negative selection (A). This result holds for simulations in which we increased the per-locus mutation rate by 2 orders of magnitude (B). SLiM3 recipes are available via Data Dryad at https://doi.org/10.5061/dryad.rfj6q577d [22]. Gene tree counts for both simulations, A and B, are available in S1 Data.
(PDF)

**S5 Fig. Present-day species distributions for 4 African Papionini (*Papio*, *Theropithecus*, *Mandrillus*, and *Cercocebus*) and 3 Asian *Macaca* species included in the introgression analysis.** The ancestral *Macaca* distribution (gray shading) is inferred from *Macaca* fossil localities in Africa and Europe as reviewed in Roos et al. [106]. The ancestral *Macaca* distribution likely represents only a fraction of the species range from the late Miocene to the late Pleistocene in Africa and Europe. The contemporary distribution of the African *Macaca sylvanus* (bright green) is included for reference; the current distribution of *Macaca nemestrina* is completely contained within that of *Macaca fascicularis*. Fossil localities for *Theropithecus* species hypothesized to overlap contemporaneously with various ancestral *Macaca* are included. Citations for spatial data of extant species: *M. nemestrina* (Richardson et al., 2008), *M. fascicularis* (Ong and Richardson, 2008), *M. sylvanus* (Butynski et al., 2008), *Macaca mulatta* (Timmins et al., 2008), *Theropithecus gelada* (Gippoliti et al., 2019), *Papio anubis* (Kingdon et al., 2008), *Cercocebus atys* (Oates et al., 2016), and *Mandrillus leucophaeus* (Oates and Butynski, 2008). Base map was obtained from the public domain map database Natural Earth (http://www.naturalearthdata.com/downloads/).
(PDF)

**S1 Table. Genomes analyzed in this study with the original NCBI release date, the publication for the reference used, and the accession number for the assembly.** When possible, the most recent version for each genome was used.
(DOCX)

**S2 Table. All published genomes used in this study, including links to the assemblies and NCBI BioProjects.** Annotation information is included for each genome at the time of download.
(XLSX)

**S3 Table. Orthogroup, protein name, human chromosome number, and coordinates for the single-copy human orthologs used in the 1,730 gene analysis.** Alignment files are named by orthogroup, allowing the use of this table to identify the protein in each alignment.
(XLSX)

**S4 Table. Gaps/ambiguities by species and as a percentage of total alignment length.** * denotes species sequenced this study.
(DOCX)

**S5 Table. Lengths for each 40-locus concatenated alignment used in the molecular dating analyses.** Each dataset was analyzed twice until node age estimates converged (15–25k steps) using a log-normal auto-correlated model [139]. Datasets are available via Data Dryad at https://doi.org/10.5061/dryad.rfj6q577d [22].
(DOCX)

**S6 Table. Fossil calibrations employed in this study.** Node numbering corresponds to the numbering in Fig 3. Median underflow/overflow for each calibration was calculated from 20 independent runs performed on 10 datasets (2 runs per dataset).
(DOCX)

**S7 Table. Mean node age for 20 independent PhyloBayes dating runs.** Node numbers correspond to the numbering in Fig 3. The 95% HPD intervals were calculated by averaging the minimum and maximum of the 95% HPD interval for each dating run. HPD, highest posterior density.
(DOCX)

**S8 Table. Quartets used to test for significant Δ values for internal branches of the primate tree.** Branches tested correspond to the labeled branches in Fig 3. After correcting for multiple comparisons (Dunn–Šidák, $P = 0.00301$), 3 internal branches and 8 quartets were found to have significant Δ values, indicating a likely introgression event.
(DOCX)

**S1 Data. The Excel workbook contains 5 different tabs.** Tab 1, Fig 3 Data: consists of the node age estimates for all 20 independent PhyloBayes dating analyses as well as the run used to determine the prior for each node; each estimate is plotted separately in Fig 3. Tab 2, Fig 3 Data for R: the same data as in tab 1, but formatted for analysis with the accompanying R script "plot_DATING.R" available via Data Dryad: https://doi.org/10.5061/dryad.rfj6q577d [22]. Tab 3, S3_Fig_Data: the data used to generate S3 Fig. The average node ages estimated in tab 1 are used here to plot age vs. concordance factors estimated for each node in IQ-TREE. Tab 4, S4_Fig_PanelA_Data: contains the tree counts that resulted from the SLiM3 simulation conditions pictured in S4A Fig. Tab 5, S4_Fig_PanelB_Data: contains the tree counts for the SLiM3 simulation conditions pictured in S4B Fig. SLiM3 recipes for both simulations are available via Data Dryad at https://doi.org/10.5061/dryad.rfj6q577d [22].
(XLSX)

## Acknowledgments

We thank Yue Liu for assistance in assembling the genomes, and Fábio Mendes and Gregg Thomas for helpful advice.

## Author Contributions

**Conceptualization:** Dan Vanderpool, Matthew W. Hahn.

**Data curation:** Daniel Hughes, Shwetha Murali, R. Alan Harris, Muthuswamy Raveendran, Donna M. Muzny, Richard A. Gibbs.

**Formal analysis:** Dan Vanderpool.

**Funding acquisition:** Jeffrey Rogers, Matthew W. Hahn.

**Investigation:** Dan Vanderpool.

**Methodology:** Dan Vanderpool, Bui Quang Minh, Robert Lanfear, Kim C. Worley, Matthew W. Hahn.

**Project administration:** Dan Vanderpool, Jeffrey Rogers, Matthew W. Hahn.

**Resources:** Daniel Hughes, Shwetha Murali, R. Alan Harris, Muthuswamy Raveendran, Donna M. Muzny, Richard A. Gibbs.

**Software:** Dan Vanderpool, Bui Quang Minh, Kim C. Worley.

**Supervision:** Jeffrey Rogers, Matthew W. Hahn.

**Visualization:** Dan Vanderpool.

**Writing – original draft:** Dan Vanderpool, Matthew W. Hahn.

**Writing – review & editing:** Dan Vanderpool, Jeffrey Rogers, Matthew W. Hahn.

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
