## [Editor Report · Decision Letter 0]

22 Apr 2020

Dear Dr Vanderpool, 

Thank you for submitting your manuscript entitled "Primate phylogenomics uncovers multiple rapid radiations and ancient interspecific introgression" for consideration as a Research Article by PLOS Biology.

Your manuscript has now been evaluated by the PLOS Biology editorial staff, as well as by an academic editor with relevant expertise, and I'm writing to let you know that we would like to send your submission out for external peer review.

Please re-submit your manuscript within two working days, i.e. by Apr 24 2020 11:59PM.

Kind regards,

Roli Roberts

Senior Editor

PLOS Biology

---

## [Decision Letter · Decision Letter 1]

6 Jun 2020

Dear Dr Vanderpool,

Thank you very much for submitting your manuscript "Primate phylogenomics uncovers multiple rapid radiations and ancient interspecific introgression" for consideration as a Research Article at PLOS Biology. Your manuscript has been evaluated by the PLOS Biology editors, an Academic Editor with relevant expertise, and by three independent reviewers. We also recruited a fourth, but they have been unable to submit in timely fashion.

You'll see that all three reviewers were very positive about your study, but each requests a number of textual and presentational changes, and in some cases some additional analyses, which should be addressed before further consideration.

In light of the reviews (below), we will not be able to accept the current version of the manuscript, but we would welcome re-submission of a much-revised version that takes into account the reviewers' comments. We cannot make any decision about publication until we have seen the revised manuscript and your response to the reviewers' comments. Your revised manuscript is also likely to be sent for further evaluation by the reviewers.

We expect to receive your revised manuscript within 2 months. 

**IMPORTANT - SUBMITTING YOUR REVISION**

*Re-submission Checklist*

*Published Peer Review*

*PLOS Data Policy*

*Blot and Gel Data Policy*

Sincerely,

Roli Roberts

Senior Editor

PLOS Biology

REVIEWERS' COMMENTS:

Reviewer #1:

[identifies himself as Matt Pennell]

This is a really exceptional contribution to phylogenetics. The analyses were well-thought and thoroughly done -- I feel very confident in your conclusions. And the writing is clear throughout; I appreciated the mix of methodological details with natural history/biological context. I anticipate that I will add this to the reading list for my phylogenetics class next year as a superb example of the power of modern data and approaches to phylogenomics. 

I have only a few minor points that I think might be worth addressing before publication. 

Line 150: I think this is slightly confusing since the underlying gene trees used in the ASTRAL analysis were also estimated with IQTREE2. The way it is written it makes it seem that IQTREE2 was used only for the concatenated analysis.

Line 163: I will admit my ignorance and say that I didn't know what the phrase "maximal local posteriors" meant and had to look it up. Given that this is a relatively new/uncommon term, I think it might be worth devoting an extra clause or sentence to explaining.

Line 167-168: I would really appreciate a few additional sentences giving an outline of this controversy. As it stands now, it is really hard to know what all the fuss is about. In a later section, you do a really great job discussing the controversies over the placement of tarsiers and I think something similar would be very useful here. 

Line 237-238: I realize that this is a throwaway line/citation but I think the evidence for the association between primate diversification and global temperature is pretty weak in my view. I guess it is fine to speculate but this is presented as a clear fact. Given how precise the rest of the paper is, this statement is uncharacteristically flippant.

Line 264: I understand what you are getting at here but I think it is worth spelling out the argument for using parsimony in this case; it is not really clear from the text alone why you think parsimony is going to be a more consistent estimator than ML here.

Line 294: Do you have any (more directa) evidence that intralocus recombination is an issue here? I think it may well be true but I think this is a stretch to infer from the increased concordance for the MP trees.

Line 310: I understand why you had to choose small subsets of the data for the dating analyses but I am wondering what the justification for selecting genes at *random*. Given all the excellent work you've done to determine the branching pattern, I am thinking that it might be worth leveraging this information to pick genes for the dating analyses to reduce the topological noise. Maybe this is a bad idea (and please tell me if it is) but if you wanted to estimate divergence times, why not select randomly from genes that matched the species topology??

Line 333: Could you please note in the MS why you didn't include most of the fossils used by Herrera et al. Were most of the ones you didn't use associated with tips rather than nodes. 

Line 418: I do not think that the line about PhyloNet adds anything to the paper. Who knows what to make of the failure to converge? 

Again, congratulations on an outstanding achievement. 

Reviewer #2:

Vanderpool et al. analyze primate genomes (including three new ones) to asses phylogenetic relationships, incomplete lineage sorting (ILS), and introgression among divergent lineages. The paper is generally well done and well written, and the conclusions generally follow from the analyses done by the authors. My primary requirement for revision is to simply post all relevant tree and data files and not just some of these, an essential revision. The impact of the work will be high I think, and otherwise, the statistical analyses are high quality, and the supplementary information and figures are fine. 

The authors also might consider the following in revising their manuscript:

1) line 85. The authors note that, " Compared to recent hybridization, introgression that occurred between two or more ancestral lineages (represented by internal branches on a phylogeny) is difficult to detect." Introgression involving completely extinct side branches in these trees might also be a problem, as is hybridization with extant lineages for which genomes have not been published. Is there any way to account for this type of introgression as well? Authors could comment on this point either way, as I fear that sometimes these types of hybridization events can get lost in the sauce of comparative genomic studies.

2) Table 1. What are the contig N50s for each assembly? The genome sequencing and assembly methods in the methods section could be fleshed out a bit more than the current terse text. It is not my specialty, but the description here seems a bit minimal, and it might be difficult for the reader to understand exactly how the genomes were assembled with long reads or without in one case, if I am reading things correctly?

3) line 188. The authors note regarding conflicts among gene trees at a node that, "...the concordance factors also indicate that a majority of individual topologies have histories that differ from the estimated species tree." But, most of this conflict, I would bet, is not due to different gene 'histories', and much of this conflict is surely due simply to gene tree reconstruction error. For example, at the node that groups the tarsier (Carlito) with Simiiformes, the authors report 60% of genes conflicting, but in analyses of retroposon insertions more than a 100 such rare genomic events support this node with absolutely zero conflict. Furthermore, simulations have shown for similar situations that there is extensive gene tree reconstruction error for fairly short internodes that are more than 50 million years old. So, I do not agree with the authors' interpretation here. The authors acknowledge this later on line 198, but I do not see any necessity in saying one thing and then ten lines later reversing course. Same for homoplasious sites versus sites that are incongruent due to ILS or introgression; it is a challenge to distinguish these from each other after a certain level of divergence or maybe even for fairly recent divergences such as chimp, human, gorilla. A recent study based on large insertions shows that such rare genomic changes completely uniformly favor the chimp+human grouping with absolutely zero conflict.

4) line 204. As noted just above, for placement of tarsier at least, the authors' argument here on biological causes of gene tree incongruence versus gene tree reconstruction error that causes conflict do not hold up from my perspective. So, I think a more cautious interpretation is warranted given the uncertainty.

5) line 216. I see here that the authors mention the position of tarsier and note that there is extensive conflict among characters and among gene trees at this node, but this is completely contrary to a published analyses of transposon insertions where more than 100 transposons uniquely support this clade with absolutely no conflicts. So, if think that the gene tree and character conflicts are actually real, this needs to be reconciled with the published transposon insertion data. I think it would be a challenge to conclude other than that: 1) the transposon data were made up and are not real (i.e., these other authors cheated), 2) the sequence data analyzed in the current manuscript are homoplasious and gene trees are inaccurately reconstructed, or 3) some sort of weird natural selection drove retroposon insertions to be fixed very rapidly relative to nucleotide substitutions at both silent third codons and selected first and second sites in protein coding genes. I think the third interpretation would be a challenge to argue for, and the first explanation imlies that someone else cheated.

6) line 222. The studies that cite conflicting previous placements of the tarsier are ones that analyzed little data. There are many previous analyses that have strongly supported tarsier plus Simiiformes based on extensive data, and I think none based on lots of data that have supported alternatives, except for analyses where researchers have botched their analyses (e.g., Song et al., 2012).

7) line 233. Change "Concatenation Affects Resolution of the New World Monkey Radiation" to "ML Concatenation Affects Resolution of the New World Monkey Radiation"? The parsimony concatenation seems to agree with the ASTRAL tree here, and only the ML concatenation tree conflicts? Also, the large ML concatenation tree of Springer et al. (2012) conflicts with the ASTRAL tree from the current study but agrees with ML concatenation analysis of the current study. Perhaps this is a bias due to ML concatenation but not concatenation generally; what does Perleman et al. ML consatenation support? If this one also supports the authors' interpretation, this further supports the discussion here. An interesting empirical result as I have seen few conflicts between ML concatenation and ASTRAL in previous published work.

8) line 258. Change "relatively low posterior support" to "very low posterior support". The PP score is really bordering on minimal?

9) line 290. The inference using parsimony gene trees and ASTRAL is novel and quite interesting. There has been an affliction in the systematics community in that workers are obsessed with ML methods due to statistical consistency, but of course, this only applies when there is no ILS and there is a single 'gene tree' for all of the data. Given that there is not unlimited data of this type in any empirical case study, it could be argued that the early simulations that showed parsimony to be lacking/inconsistent are, in the end, bogus and irrelevant. It is not the first time that arrogance of evolutionary modelers has led to mass confusion in the field. I fear that the same thing is currently happening on a much grander scale in ecological modeling of COVID death count predictions, but in this latter example, we are dealing with actual life and death. Sad... But, it is nice that the authors here have actually used a simulation result from the Hahn group and taken the time to run the parsimony trees to reveal this interesting ASTRAL pattern.

10) line 294. Change " biases of concatenation" to " biases of ML concatenation for gene sub-segments with conflicting histories".

11) line 358 and above paragraphs. I found the discussion of divergence times to be reasonable, even-handed, and well stated. The authors discuss most of the critical issues that impact estimation of dates such as these and explain possible reasons for discrepancies with previous work.

12) line 370. The authors note that it is a challenge to infer introgression at deep nodes due to multiple hits. This is true, but recently Springer et al. have argued that this is a rationale for instead using transposon insertion characters for this purpose, given that these characters are generally thought to be lower-homoplay characters based on their mode of mutation. Have the authors attempted to used these characters for inferring gene flow among divergent lineages? If there are enough such characters, I would expect these data to be much more effective than the strategy taken by the authors who used gene trees, as opposed to individual SNPs. For using gene trees, the challenge is always to determine recombination break points accurately and to deal with different expected average sizes of coalescence-genes for gene trees that support the species tree vs. gene trees that conflict with the species tree (my understanding is that the latter gene trees are generally expected to be shorter). It perhaps goes beyond the scope of the current paper to score transposon insertions, but since genomes are assembled, this would not be too much work to do? If not, it might be worthwhile to at least suggest this strategy for future work in their discussion of this issue, as I think this is the most productive way forward, especially for divergences that are even older, where even more multiple hits are expected and gene tree reconstruction becomes even more challenging.

13) line 376. I am guessing that lack of usage of this test is in large part due to the fact that 'genes' like the ones used in this study often span multiple recombination units, so the protein-coding exons strung together here do not likely represent single gene tree histories, but instead many. The authors acknowledge this somewhat, but I think it is a bigger problem than they maybe let on here?

14) line 418. In addition to PhyloNet, Cecile Ane's group (and others) have developed methods in which introgression pathways are allowed in an ASTRAL-like quartet approach to the ILS+gene flow problem. Might the authors attempt an analysis such as this that accounts for both ILS and gene flow to infer a network using an actual optimality approach (as opposed to PhyloNet which remains a bit mysterious to me in terms of how and why and what it spits out at the end of the process)?

15) line 450. In terms of caveats, I see that the authors stress more here the issue of recombination, population subdivision, etc., but there is also the problematic aspects of negative selection, which impacts neutral MSC assumptions, in particular because different protein-coding loci are likely under a variety of selective constraints, as well as selective sweeps and diversifying selection. Proponents of the coalescence approach have tried to avoid these hard truths, but we are starting to see some movement on this front after a recent (and ongoing) exchange in the literature. For example, He et al. (2019: MBE; "Asymmetric Distribution of Gene Trees Can Arise under Purifying Selection If Differences in Population Size Exist"), but I think the situation is even more dire than this publication lets on... I predict that it will become very clear that negative selection, of various degrees, will be very problematic for MSC inference and interpretation as time moves on (as argued in Springer and Gatesy, 2016, but denied by "leaders in our field" in Edwards et al., 2016). Sometimes sets of leaders are really just followers, perhaps?

16) line 476. Again here, as long as there are enough active retroelements, I think using these to infer gene flow of ancient lineages solves this problem?

17) line 488. The authors conclude that, "Our analyses reveal how concatenation of genes or even of exons can mislead maximum likelihood phylogenetic inference in the presence of discordance, but also how to overcome the biases introduced by concatenation in some cases." I am not necessarily convinced that concatenation has failed in any way in this study relative to coalescence methods such as ASTRAL. A particular brand of this format, ML concatenation, might have failed?

18) line 545. The authors note that for concatenation, they did the following: "Q-TREE v2-rc1 was used with all 1,730 aligned loci to estimate a maximum likelihood concatenated (ML-CONCAT) tree with an edge-linked, proportional-partition model". It might be best for those who have not used IQ-TREE to describe exactly what is entailed in such an analysis. Was a single substitution model used for the entire concatenation, or were separate substitution models used for each gene in the overall alignment as in the coalescence analyses? Was just one set of branch lengths used, or were different genes allowed to have different branch lengths? Were different codon positions given different branch lengths and/or substitution models?

19) line 575. Why just use four taxa here. This would seem to increase the probability of long branch artifacts as opposed to more complete sampling of taxa that will subdivide long branches? This type of sampling might in fact lead to asymmetry in results, rather than provide information on that asymmetry? How do results, for asymmetry, differ when including all taxa (as in the initial estimates that focused on seven nodes) versus the redos with just four taxa that the authors seem to prefer? Using four taxa automatically gets rid of conflict by removing lineages perhaps and increases the number of relevant loci, but perhaps this is sort of sweeping real issues under the rug?

20) line 584. Given that the authors note much evidence for introgression and also argue that concatenation can be misleading, it is potentially problematic that the clock analyses are all based on concatenation, which would seem to offer distortions. The clock methods used do not take extensive introgression or ILS into account. Perhaps the authors should qualify that ILS and introgression could cause distortions in their concatenated clock analyses (perhaps too old dates due to deep coalescence or perhaps too shallow dates due to introgression, etc.). The authors could run *BEAST which takes ILS into account to get divergence times, as the authors think this is a concern, but note that this coalescence program, like all coalescence methods, are also deeply impacted by introgression (just like concatenation). The authors naively seem to think that introgression does not impact standard coalescence methods, but this is certainly not the case, in particular for what are considered by some the best methods (i.e., *BEAST, which has a least common denominator problem with outlier genes as noted in the original description of the method). The authors should also note how genes were modeled in the concatenated IQ-TREE analyses relative to the concatenated clock analyses. Again, was each gene given a different model or one model for all genes?

21) line 608. The authors note that, " Single-copy ortholog alignments of CDS sequences produced using Guidance2 and the Nexus formatted dated species tree phylogeny are available through DataDryad (https://doi.org/10.5061/dryad.rfj6q577d)." The authors should also provide, optimal gene trees, bootstrap consensus gene trees, and concatenated matrixes (with gene partitions) used in their IQ-TREE and clock analyses so that their analyses can be replicated by others if need be. This is perhaps my strongest comment on revision. Hard to tell what was done unless all of this material is posted for other scientists to see. So, this revision is essential from my perspective.

Reviewer #3:

The manuscript submitted by Vanderpool et al. describes the generation of three novel primate genome sequences, the mining of orthologous protein coding genes across 23 published primate genomes and several outgroup genomes, and an in-depth phylogenomic analysis. The authors estimate the phylogeny and divergence times of primates and compare their results to those of previous studies. The methods and analyses are, for the most part, sound. The results confirm relationships identified in many previous studies on primate phylogeny. The real novelty in this manuscript is the detailed analysis of genealogical discordance and evidence for ILS and ancient hybridization within several lineages of primates. This section of the manuscript is quite strong and I found the analyses and interpretations to be well-reasoned and sound, and provide important insights into this aspect of early primate evolutionary history. I believe that the findings are very interesting and publishable, but there are a number of aspects of the analysis that could be improved, which I believe will lend greater confidence to their conclusions.

1. The new reference genomes are fairly phylogenetically restricted. It might help the authors message if there is some mention as to why the addition of these particular taxa are important or strategic from the standpoint of phylogenetic resolution and hybridization.

2. On page 5, it would be helpful to mention the type of sequencing platforms used for each genome.

3. Given the availability of whole genome sequences from so many primates, I was disappointed that the phylogenetic analysis was restricted to only 1,730 single-copy genes, and did not incorporate non-coding sequence alignments. In many cases they could have better ruled out gene tree reconstruction error if the phylogenetic information content of each locus was increased. For example, primate monophyly is supported for only 55% of the gene trees, which in most cases is probably due to lack of phylogenetically informative sites and probable rooting errors. Does gene tree discordance from the assumed species tree correlated with cds alignment length? Also, it would have allowed for an assessment of discordance across a more significant majority of the genome, rather than a small proportion of coding sites. In my opinion, the quality of this report would be markedly improved if the scope of the analysis was expanded beyond protein coding loci to include intronic sequences and the alignments based on genomic sequences rather than inferred isoforms, which are prone to annotation errors.

Also, I feel that the authors have missed an opportunity to catalog rare genomic changes (insertion and deletion events) such as retroelement insertions. Because these events occur only once at a single location they are not prone to within locus recombination, and can be analyzed with D statistics. The addition of these types of analyses, particularly with regard to the basal relationships within Cebidae and Colobinae, would strengthen the results.

4. Some important details for the 1,730 loci are missing, such as the chromosome distribution of the loci, whether any are tightly linked, and the breakdown between autosomes and sex chromosomes. Perhaps add a table with this information.

5. I did not see any methods or details surrounding filtration of dubious protein coding sequence alignment regions, given that the gene set is based on the longest isoform from each species, and the ends of isoforms are often poorly annotated. This step is fairly important for the analysis of deeper evolutionary divergences, where alignment anomalies are commonplace and methods have been described to deal with these issues (see Liu et al. PNAS [2017], Mason et al. Sci Adv. [2016], Springer & Gatesy, Systematics and Biodiv. [2018]). Some reporting of these details would be important to have confidence in the assessment of gene and sitewise discordance estimates, and that these values aren't unnecessarily inflated due to poor alignment quality and exon, isoform paralogy. For example, on pgs. 9-10 (lines 199-201) the authors correctly point out the various technical problems (including poorly aligned sequences, etc.) that can result in gene tree error, but they don't provide any in-depth analysis of their own data set in this regard to determine how often gene tree error can be attributed to these different factors.

6. The use of a single outgroup (and most notably the mouse, which is extremely accelerated in terms of nucleotide substitution rate compared to all other placental mammals) is potentially problematic and may have led to long branch attraction and other artifacts that would increase the amount of discordance pertaining to the relationships between primates, colugos and treeshrews. The authors should add at least one lagomorph; the rabbit and pika genomes are of similar quality and annotation as the primate genomes. Also, in mentioning previous literature, the authors might note that studies that used rare genomic changes (indels and retroelement insertions) produced statistically significant results that are not prone to the same bootstrap issues raised by the authors on lines 171-174. **Minor point, the word "probability" is missing after "posterior" on line 172.

7. Claiming parsimony (pg. 13) would not produce a biased topology with concatenation methods seems a bit of an overstatement. Parsimony methods are more prone to long branch attraction artifacts than ML methods. The authors should consider qualifying their comment here. 

8. Line 391, "then re-aligned orthologs present in a single copy in each taxon"…please clarify why this was done.

---

## [Decision Letter · Decision Letter 2]

29 Sep 2020

Dear Dr Vanderpool,

Thank you for submitting your revised Research Article entitled "Primate phylogenomics uncovers multiple rapid radiations and ancient interspecific introgression" for publication in PLOS Biology. I have now obtained advice from two of the original reviewers and have discussed their comments with the Academic Editor. 

Based on the reviews, we will probably accept this manuscript for publication, assuming that you will modify the manuscript to address the remaining points raised by the reviewers. Please also make sure to address the data and other policy-related requests noted at the end of this email.

IMPORTANT:

a) Please attend to the outstanding requests from rev #2.

b) Please address my Data Policy requests (further down letter).

c) You currently state that an ethics statement is not needed. However, we note that you used samples of primate heart and blood tissue provided by San Diego Zoo and Washington National Primate Research Center. Please could you provide details of the terms and approvals under which those samples were obtained?

We expect to receive your revised manuscript within two weeks. Your revisions should address the specific points made by each reviewer. In addition to the remaining revisions and before we will be able to formally accept your manuscript and consider it "in press", we also need to ensure that your article conforms to our guidelines. A member of our team will be in touch shortly with a set of requests. As we can't proceed until these requirements are met, your swift response will help prevent delays to publication.

- a cover letter that should detail your responses to any editorial requests, if applicable

*Copyediting*

*Published Peer Review History*

*Early Version*

Sincerely,

Roli Roberts

Senior Editor,

rroberts@plos.org,

PLOS Biology

ETHICS STATEMENT:

-- Please include the full name of the IACUC/ethics committee that reviewed and approved the animal care and use protocol/permit/project license. Please also include an approval number.

-- Please include the specific national or international regulations/guidelines to which your animal care and use protocol adhered. Please note that institutional or accreditation organization guidelines (such as AAALAC) do not meet this requirement.

-- Please include information about the form of consent (written/oral) given for research involving human participants. All research involving human participants must have been approved by the authors' Institutional Review Board (IRB) or an equivalent committee, and all clinical investigation must have been conducted according to the principles expressed in the Declaration of Helsinki.

DATA POLICY:

Many thanks for providing raw data, alignments and trees in NCBI BioProject and Dryad. However, we also that all individual quantitative observations that underlie the data summarized in the figures and results of your paper be made available in one of the following forms:

Regardless of the method selected, please ensure that you provide the individual numerical values that underlie the summary data displayed in the following figure panels as they are essential for readers to assess your analysis and to reproduce it: Figs 3, S3, S4. NOTE: the numerical data provided should include all replicates AND the way in which the plotted mean and errors were derived (it should not present only the mean/average values).

Please also ensure that figure legends in your manuscript include information on where the underlying data can be found (e.g. supplementary file, Dryad, etc.), and ensure your supplemental data file/s has a legend.

REVIEWERS' COMMENTS:

Reviewer #2:

This is a re-review by reviewer #2, so I will mainly comment on issues that I brought up in the first round. Generally, the authors responded to nearly all of my queries with thoughtful responses and edits to their paper, and they included a more than required response to my query about negative selection by completely redoing a published simulation study that they found lacking. My only two remaining requests have to do with publication of alignments for both genes and transposon insertions. 

The authors note that they have posted their trimmed DNA sequence alignments, and they have clarified how this trimming was done. However, for other researchers to exactly follow what was originally aligned and then tossed from each alignment is critical for interpretation of their results and conclusions. This is because many of their conclusions have to do with recombination and possible gene flow. These inferences derive from conflicts among loci and within loci that are caused by these processes. So, in order to track what is what in their overall analysis, the precise homology relationships of sequences should be presented both in the original alignments of what the authors thought are homologous regions of each species' genome (to see whether initial annotations of genes are justified) and also in the final alignments that were used in analyses (which the authors have posted already). 

Second, I think it would be important to post a subset of the alignments for transposon insertions for the position of the tarsier relative to bushbaby and human. This would be an important contribution, because as far as I know, no shared derived transposons have been documented for the human+bushbaby clade in the literature, but the authors have discovered possibly as many as 341. Hartig et al. (2013) instead found 104 transposon insertions that cleanly support a human+tarsier clade with no conflicts at this node. To be blunt, I do not really believe the new result that support for this latter clade (435 transposons) is countered by extreme conflict (341 transposons), because the authors have not looked at the transposon insertion site alignments in detail to assess the quality of the alignments as done in Hartig et al. Although, I do not expect the authors to go through all of these 341 conflicting transposon alignments as this is not the primary focus of the paper, they should check a subset of these to be sure that at least a good number of these are convincing in terms of alignment ends, insertion points, homology of the transposon insert, etc. which can only be done currently, I think, by looking at these by eye. Ideally, it would be great if they could document 10-20 'clean'and convincing transposon insertions for the human+bushbaby clade and present these alignments in a supplementary file that would document these to skeptics (like me...). 

Aside from these two points, which can be easily addressed without so much new work, I commend the authors on their nice phylogenomic study and in their thoroughness in dealing with reviewers' queries. 

Reviewer #3:

I thank the authors for a very detailed and thoughtful response to my previous comments. I am satisfied with the revisions, and I have no further concerns.

---

## [Editor Report · Decision Letter 3]

2 Nov 2020

Dear Dr Vanderpool,

On behalf of my colleagues and the Academic Editor, Chris D Jiggins, I am pleased to inform you that we will be delighted to publish your Research Article in PLOS Biology. 

PRODUCTION PROCESS

Before publication you will see the copyedited word document (within 5 business days) and a PDF proof shortly after that. The copyeditor will be in touch shortly before sending you the copyedited Word document. We will make some revisions at copyediting stage to conform to our general style, and for clarification. When you receive this version you should check and revise it very carefully, including figures, tables, references, and supporting information, because corrections at the next stage (proofs) will be strictly limited to (1) errors in author names or affiliations, (2) errors of scientific fact that would cause misunderstandings to readers, and (3) printer's (introduced) errors. Please return the copyedited file within 2 business days in order to ensure timely delivery of the PDF proof. 

If you are likely to be away when either this document or the proof is sent, please ensure we have contact information of a second person, as we will need you to respond quickly at each point. Given the disruptions resulting from the ongoing COVID-19 pandemic, there may be delays in the production process. We apologise in advance for any inconvenience caused and will do our best to minimize impact as far as possible.

EARLY VERSION

PRESS 

Kind regards,

Alice Musson

Publishing Editor, 

PLOS Biology

on behalf of

Roland Roberts,

Senior Editor

PLOS Biology